# Neurocognitive and Neuropsychiatric Implications of Fibrosing Interstitial Lung Diseases

**DOI:** 10.3390/biomedicines12112572

**Published:** 2024-11-10

**Authors:** Zsolt Vastag, Emanuela Tudorache, Daniel Traila, Ovidiu Fira-Mladinescu, Monica Steluta Marc, Cristian Oancea, Elena Cecilia Rosca

**Affiliations:** 1Center for Research and Innovation in Personalised Medicine of Respiratory Diseases (CRIPMRD), “Victor Babes” University of Medicine and Pharmacy, Eftimie Murgu Square 2, 300041 Timisoara, Romania; zsolt.vastag@umft.ro (Z.V.); traila.daniel@umft.ro (D.T.); mladinescu@umft.ro (O.F.-M.); marc.monica@umft.ro (M.S.M.); oancea@umft.ro (C.O.); 2Department of Internal Medicine, Discipline of Clinical Practical Skills, “Victor Babes” University of Medicine and Pharmacy, No. 2 Eftimie Murgu Square, 300041 Timisoara, Romania; 3Doctoral School, “Victor Babes” University of Medicine and Pharmacy, 300041 Timisoara, Romania; 4Pulmonology Department, “Victor Babes” University of Medicine and Pharmacy, Eftimie Murgu Square 2, 300041 Timișoara, Romania; 5Department of Neurology, “Victor Babes” University of Medicine and Pharmacy, 300041 Timisoara, Romania; rosca.elena@umft.ro; 6Department of Neurology, Clinical Emergency County Hospital Timisoara, 300736 Timisoara, Romania

**Keywords:** cognitive impairment, depression, anxiety, idiopathic pulmonary fibrosis, sarcoidosis, connective tissue diseases

## Abstract

Patients with interstitial lung diseases (ILDs) associate a large variety of comorbidities that have a significant impact on their clinical outcomes and survival. Among these comorbidities is neurological impairment. This review highlights what is known about the cognitive function, central nervous system (CNS), depression, and anxiety in patients with specific forms of fibrosing ILDs, such as idiopathic pulmonary fibrosis, sarcoidosis, hypersensitivity pneumonitis, connective tissue diseases, etc. The most common pathogenic mechanisms for neurocognitive dysfunction as well as the screening methods and tools for their identification are also described in this review.

## 1. Introduction

Diffuse parenchymal lung diseases have a wide range of causes and include a significant number of conditions whose manifestations, most of the time, have an unpredictable evolution. Their pathological abnormalities predominate in the pulmonary interstitium, and fibrosis is one of the quasi-present elements; therefore, these disorders are called interstitial lung diseases (ILDs). Several known factors can lead to the appearance of interstitial damage, such as genetic predisposition, smoking, some drugs, connective tissue diseases and vasculitis, various inhaled substances (repeated exposure to certain organic particles), irradiation, malignancy, etc. In other situations, the factor that induced the production of interstitial lesions is not identified, so these forms of ILDs are classified as idiopathic [1].

Within ILDs, idiopathic pulmonary fibrosis (IPF) is the archetypal and the most studied entity. IPF is characterized by a pathological and imaging pattern known as usual interstitial pneumonia (UIP) without a known cause or associated pathology being identified [2].

Fibrosis is one of the physical–pathological phenomena that occur after an injury, through which the body tries to stop the evolution and/or to restore the respective tissue. More than 200 pulmonary fibrogenic entities have been identified [1]. However, the notion of ILD excludes diseases with mutilating and/or localized fibrosis (fibrothoracic, post-traumatic, post-suppurative, etc.)—referring only to diffuse diseases of the interstitial/parenchymal compartment.

The current concept regarding fibrosis is very dynamic; the fibrogenic process can be in one of the following situations: it can progress, remain stable, or even regress. This different manifestation of the fibrogenic process led to the delimitation of the so-called progressive fibrotic phenotypes. A progressive character always characterizes IPF. The percentage of patients with non-IPF ILDs who may develop a progressive fibrotic pattern is difficult to estimate, but it is assumed to represent 18–32% of ILDs [3].

On the other hand, independent of any disease, physiological aging of the lungs is associated with anatomical and functional changes, including a decrease in elastic recoil (in both alveoli and airways), in chest wall compliance, in respiratory muscle strength, and in the activity of the central nervous system and alveolar surface. Also, there are changes in the pulmonary matrix composition and fewer capillaries per alveoli [4].

Furthermore, comorbidities in IPF are common in older people, and IPF is considered a pathology of older adults. Researchers reported that 65% of people between 65–84 years old have at least two associated chronic diseases, and for those over 85 years old, the rate exceeds 80% [5].

The number of comorbidities and their management have been proven to significantly impact these individuals’ clinical outcomes and survival [6,7]. Among these comorbidities is neurological impairment. Although the cognitive status of patients was investigated in some specific forms of ILD, a clear neurobiological model was not proposed regarding the functionality of the brain–lung neuropsychological axis in people with interstitial lung disease [8].

Among subjects with chronic respiratory diseases, cognitive impairment has been most studied and documented in patients diagnosed with chronic obstructive pulmonary disease (COPD), where it was found to be common, being reported in both chronic hypoxia and non-hypoxic patients [9,10]. Still, in other diseases often associated with rapid and frequent development of hypoxemia such as ILDs, it is surprising that cognitive function has been rarely investigated.

Cognitive impairment, depression, and anxiety are a reality and should be a significant concern in chronic respiratory diseases, as they can cause functional disability and poor clinical outcome and treatment adherence, all of which lead to a decreased quality of life and social isolation.

The current review highlights what is known until now about cognitive dysfunction, central nervous system impairment, anxiety, and depression in patients with the most common types of fibrosing ILDs, mainly since this topic has not been sufficiently studied yet.

## 2. Methodology

We performed a comprehensive literature search in the PubMed database to identify relevant studies describing the impact of ILDs on several neuropsychiatric domains from inception to March 2024. A combination of search terms, “interstitial lung disease”, “pulmonary fibrosis”, “IPF”, “sarcoidosis”, “hypersensitivity pneumonitis”, “connective tissue diseases”, “systemic lupus erythematous”, “rheumatoid arthritis”, “scleroderma”, “Sjogren syndrome”, “polymyositis”, “dermatomyositis”, “vasculitis”, “nonspecific interstitial pneumonia”, “unclassifiable ILDs”, “neurologic”, “cognitive”, “central nervous system”, “anxiety”, and “depression”, were used to identify English language studies that had the key search terms in their title or abstract. In our review, we included original research on neurological and psychiatric manifestations in adult subjects with ILDs. The scoping search identified 5153 articles, of which 74 were directly related to our research topic.

## 3. Epidemiology of ILDs

Regarding the epidemiological data of ILDs, there are varied figures in different studies, depending on the country and ILD type. There are few studies that have described the global impact of ILDs [11]. Worldwide, the reported prevalence ranges from 6.3 to 71 per 100,000 people, and the incidence of ILDs ranges between 1 and 31.5 per 100,000 person-year [12].

The incidence and prevalence of progressive-fibrosing ILDs ranged between 6.9 and 78.0/100,000 persons, respectively 2.1–14.5/100,000 person-year [13].

The mortality rates/year for ILDs in the USA are approximately 7.3/100,000 in males and 5.9/100,000 in females. Moreover, despite more complex investigations and new therapies, an increase in the death rates has been registered in the last decade compared with approximately 18.1% [14].

The mortality varies with ILD type, progressive-fibrosing phenotype, age, sex, and comorbidities. For example, the 5-year mortality is 44.1–78% for IPF, 30–54% for unclassifiable ILDs, 35–39% for rheumatoid arthritis-ILD, 10–18% for scleroderma (SSc-ILD), and 12–16% for progressive systemic sclerosis (pSS-ILD) [15,16,17,18,19,20,21].

The wide variation in ILD prevalence reported in the literature can be explained by several factors, such as different diagnostic criteria, terminology, and classification, which have changed over the years; most studies had relatively small sample sizes; they used different study designs (prospective vs. retrospective); and some of them had insufficient follow-up time or loss from follow-up, which could clarify several cases initially labeled as working diagnosis or unclassifiable disease.

## 4. Diagnosis and Classification of Interstitial Lung Diseases

The classification of ILDs has undergone several changes over time. They were initially classified only according to the pathological characteristics until computerized tomography was available worldwide. The introduction of multidisciplinary discussion in the diagnostic algorithm, which includes the radiologist, the ILD clinician, the pathologist, and sometimes the rheumatologist or thoracic surgeon, was a significant step forward in the evolution of ILD classification [22,23].

The current classification combines clinical, histological, and imaging patterns for the final diagnosis [23]. The tools to establish the diagnosis are high-resolution computed tomography (HRCT); functional respiratory tests, including diffusing capacity for carbon monoxide (DLCo); broncho-alveolar lavage; and, sometimes, surgical biopsy/cryo biopsy [22].

Furthermore, there are several biological markers, such as anti-nuclear antibodies (ANA) profile, anti-neutrophil cytoplasmic antibody (ANCA), angiotensin-converting enzymes, allergen-specific IgGs, etc., which are used to guide the diagnosis towards a particular type of ILD. The scientific community makes considerable efforts to identify specific features and serological/genetic markers for a more precise classification of each case, a more accurate prognosis, and a better interpretation of treatment response [23,24].

Using all these tools, ILDs are currently classified into six categories: idiopathic interstitial pneumonia, autoimmune-ILD, exposure-related ILD, ILD with cysts and/or airspace filling, sarcoidosis, and other ILDs. Except for sarcoidosis, each ILD category includes several specific diseases. On the other hand, each subtype has a different potential to manifest a progressive-fibrosing phenotype [3].

## 5. The Spectrum of Neurological Impairment in Fibrosing ILDs

In the studies carried out in patients with chronic lung diseases, several neuropsychological disorders were identified, among which the most frequent was the impairment of cognitive functions [8,9,25,26,27,28,29,30,31]. However, until now, research has not yet established a clear pattern regarding the neurological damage in fibrosing ILDs [8,32].

Cognition is the most studied neurological function among ILD patients, and impairments can interfere at any of its levels: visuospatial level, memory, attention, execution, language, learning, abstraction, orientation, etc. [10].

Few studies assessed the cognitive function in particular forms of ILD, such as IPF, sarcoidosis, connective tissue diseases, or vasculitis. In patients with severe IPF, a marked decrease in cognitive function was observed compared to those in the initial stages of the lung disease [8]. Elfferich et al. revealed that cognitive impairment is a problem in people with sarcoidosis, being also influenced by disease duration and the association with depression [33].

Along with the cognitive disorders, this population faces certain functional limitations, certain emotional disturbances, and a decrease in the quality of life both for them and for their caregivers [32]. Ryerson et al. showed in their study that depression is a common comorbidity in patients with ILD, with an average of 21% being diagnosed with a clinically significant form. In IPF, the prevalence reached 24%, while in other ILDs, it was 19%. However, the severity of depressive symptoms was similar among the different types of ILD [34]. Another study indicates that the prevalence of clinically significant depression and anxiety in patients with ILD is between 7 and 12% and significantly reduces their quality of life [35].

## 6. Pathogenesis of Neurological Damage in Fibrosing ILD

Regarding the pathogenesis of other neurological diseases, research is still needed to demonstrate the relationship between fibrosing ILDs and neurological disorders in more detail. This relationship was initially studied only for the terminal stages of lung disease. Headaches, tremors, drowsiness, intracranial hypertension, loss of visual acuity, and papillary edema have been reported. Along with these, anxiety, irritability, and confusion were highlighted [36].

An important characteristic of patients with fibrosing ILD, especially those with IPF, is progressive dyspnea that leads to a marked limitation of physical activities and, over time, an increased dependence on those around them, even for simple tasks. Due to these lifestyle changes, many of the patients develop symptoms such as anxiety and depression, which significantly reduce their quality of life [37]. Depression is also associated with certain variables, such as poor quality of sleep or chronic pain. Taking into consideration ILD features and depression contributing factors, patients must also receive an antidepressant treatment, as it is unlikely to resolve without it [34].

Research from the last decades has highlighted the occurrence of cognitive deficits caused by hypoxia and/or hypercapnia. Fegyveres showed that chronic lung diseases associated with hypoxemia led to subcortical cognitive impairment and reduced attention and executive functions. A slowing of mental processing speed was also detected in hypoxic patients by performing the Trail Making Test (TMT) and Time Reaction Tests (TRT) [36]. Studies indicate that chronic hypoxia can lead to ischemic brain damage and neurodegeneration. These studies have succeeded in demonstrating that those with chronic hypoxemia and altered pulmonary function can present lesions in the white matter and even lacunar infarcts [38].

However, Sharp et al. showed that even patients with normal oxygen levels can present a mild cognitive dysfunction [39]. There are common risk factors for developing cognitive impairment or other neurological disorders in this population besides hypoxemia, such as the chronic evolution of the diseases, the phenomenon of aging, smoking status, anxiety, and depression [10].

Also, ILD patients have restrictive ventilatory dysfunction, which has been associated with a higher incidence of subclinical atherosclerosis, diabetes, and an increased risk of cardiovascular events. These pathologies, in turn, can increase the risk of developing dementia and cognitive disorders. The pro-inflammatory status that develops as a consequence of altered pulmonary function can also trigger them. Moreover, studies showed an increased risk of dementia in patients with reduced lung function and elevated levels of C-reactive protein [37,39,40,41].

On the other hand, even in individuals with normal cognitive function but poor motor control, movement can be difficult [12]. Along with the occurrence of neurological impairment in patients with ILD, the limitation of physical activities may also emerge because the initiation of limb movements occurs through cortical activation [42]. Besides this, during deep breathing, the function of the respiratory muscles also requires cortical activation. Therefore, a neurological disorder can make the patient’s physical activity difficult and reduce their quality of life [12].

Other factors that have a negative impact on the quality of life of patients with ILDs, along with cognitive dysfunction, are the progression of dyspnea, persistent or night cough, sleep apnea, and an increased number of medications [37]. The pathogenesis of neurocognitive and neuropsychiatric implications in fibrosing ILD is summarized in Figure 1.

## 7. Neurological Impairment in the Most Common Types of Fibrosing ILD

### 7.1. Neurological Dysfunction in IPF

Comorbidities play an essential role in IPF heterogeneity. A high rate of comorbidities has been documented for IPF, which can be represented by the so-called “IPF comorbidome”. The prevalence of these comorbidities and the strength of their association have a great impact on IPF morbidity and mortality. The comorbidome could predict the prognosis for individual patients with IPF and, thus, improve personalized treatment [7].

Recent research has assessed patients with IPF from a comorbidities point of view, showing that only 12% had no comorbidities. In comparison, 58% presented three comorbidities, and 30% of the patients had between four and seven comorbidities. Comorbidities can be divided into respiratory and non-respiratory. Among the respiratory ones, the most common are sleep apnea, emphysema, lung cancer, and pulmonary hypertension. At the same time, the common non-respiratory ones are cardiovascular, gastro-esophageal reflux, diabetes, and depression [10].

#### 7.1.1. Cognitive Impairment in IPF

There are few studies that assessed the impact of IPF on neurological function, with most of them focusing on cognitive impairment. In a center where patients with a severe form of IPF were examined, researchers observed that they had difficulties in understanding and recalling some aspects of their disease. The neurologic function was assessed through different tests: the Grooved Pegboard test for psychomotor speed; the Sroop 3 test that analyzes processing speed; TMT B, which measures the speed of divided attention; and BNT for confrontation naming. They concluded that patients with a severe form of IPF presented lower scores in these tests than those with a mild–moderate form of IPF or the control group [32].

Another study analyzed the cognitive function of patients with IPF and COPD through the Montreal Cognitive Assessment (MoCA) test. The scores obtained were significantly lower for participants with COPD or IPF than those in the control group. Cognitive function in patients with COPD and IPF was at least mildly affected, especially for domains such as language abilities, visuospatial function, and delayed recall. However, they did not find a correlation between demographic data, anxiety or depression, and cognitive function in patients with IPF [10]. Another cross-sectional study on patients with IPF also used the MoCA as a tool to detect cognitive dysfunction. The results also showed that subjects with IPF presented a mild cognitive disorder in terms of working memory, language, and visuospatial skills [10].

On the other hand, a prospective, observational study in which 30 patients with IPF with oxygen saturations within normal limits were included examined the cognitive function of these patients, comparing it with a group of patients with COPD and a control group who were smokers. They showed that almost 50% of the patients with IPF had a mild cognitive disorder, which could not be explained by their age [38].

#### 7.1.2. Anxiety and Depression in IPF

Several studies showed that symptoms of depression and anxiety are more frequent in IPF patients compared with age-matched control groups [43]. One study showed that the prevalence of anxiety can be up to 60%, and depression varies between 24.3–49.2% in this population [44,45,46]. De Vries et al. highlighted the fact that significant depressive symptoms were reported in approximately 25% of patients with IPF [43]. At the same time, in another study, the prevalence of sadness or depression was 18.2%, and anxiety was 15.9%. The large variability in the prevalence of these comorbidities could be explained by the fact that they used different assessment methods [45].

Furthermore, in severe, progressive forms of IPF, the presence of anxiety or depression was reported more frequently. Patients with significant respiratory distress, intense chronic cough, or severe dyspnea more often exhibit symptoms of depression [34,46,47,48].

A retrospective study in which 121 newly diagnosed IPF patients were included analyzed the impact of depression on these patients. They assessed this population by several tools: oxygen pressure at rest, Hospital Anxiety and Depression Scale (HADS), St. George’s Respiratory Questionnaire (SGRQ), 6 min walk test (6MWT), Baseline Dyspnea Index (BDI), and pulmonary function tests. The study concluded that depression is a factor that significantly influences the health-related quality of life, and it must be effectively managed [47].

#### 7.1.3. Quality of Life (QOL) in IPF

The relation of depressive symptoms and dyspnea with the QOL in IPF patients was also investigated. De Vries et al. showed that, for subjects with the same demographic characteristics, the QOL of IPF patients was primarily impaired by their lack of independence and poor physical health compared to the control group. A decreased pulmonary function is associated with low QOL and higher depression scores [49,50].

Even if anxiety can also be a contributing factor to a poor quality of life, there is still not enough research on the anxiety–IPF association [46].

Nevertheless, studies have shown that, although depression and anxiety are more common in patients with IPF, they do not have an impact on mortality or exacerbations but significantly influence their QOL, and for this reason, treatment in this direction is necessary [46]. Decreased QOL, severe dyspnea, onset of depression, and anxiety are additional sufferings for patients with IPF [44,49]. Therefore, it has been observed that patients with IPF who have these ailments show lower adherence to treatments, thus requiring greater social and medical support [51].

Regarding other neurological disorders (nonpsychotic mental disorders, extrapyramidal and movement disorders, epilepsy, nerve, nerve root and plexus disorders, etc.) in patients with IPF, there have been no data in this research direction. Therefore, further studies are needed to better understand the connections between IPF and neurological impairment.

### 7.2. Neurological Dysfunction in Sarcoidosis

Regarding sarcoidosis, some studies have shown that cognitive disorders are a significant problem for these patients, especially because sarcoidosis mainly affects young adults. Elfferich et al. analyzed whether the presence of fatigue, small-fiber neuropathy, and depression is related to a higher frequency of cognitive disorders or if there is an unknown mechanism that unites these symptoms with those of cognitive deficit. The results showed that healthy subjects have a lower risk of cognitive disorders than patients with sarcoidosis. Over one-third of the subjects with sarcoidosis had cognitive impairment compared to only 14.3% of the healthy individuals in the control group. Therefore, everyday cognitive deficits in patients with sarcoidosis have been shown to be significant [33,52]. However, other research showed no relationship between the Cognitive Failures Questionnaire (CFQ) scores and certain patient parameters, such as pulmonary function, radiological stage of sarcoidosis, inflammatory tests, dyspnea, or sleep disorders [53]. Omcikus et al. have concluded that cognitive dysfunction is more often encountered in patients with sarcoidosis than in age-matched control groups and depends on the duration of the disease or the onset of depression [8,53].

Patients suffering from neurosarcoidosis have a particular aspect. It should be emphasized that neurosarcoidosis is often not initially recognized [54]. The central nervous system in patients with sarcoidosis is affected between 5 and 10%; the hypothalamus, cranial nerves, and pituitary glands are most often involved [43,55]. In a study in which 121 patients were included, fatigue was present in 92.4%, and 53.4% reported extreme fatigue. These patients were also examined for their daily cognitive function compared with individuals from the general population. The results showed that everyday cognitive dysfunction represented a particular problem in patients with neurosarcoidosis. Comparing the two groups, it was shown that approximately one-third of the general population presented cognitive disorders, while more than half of the patients with neurosarcoidosis had these deficits. The study also highlighted that cognitive failure can be a predictor of its occurrence, the most important factors being the symptoms of small-fiber neuropathy (SFN) and fatigue [53].

Another study in which 315 patients with sarcoidosis were included analyzed the impact of self-reported cognitive disorders. It was found that cognitive difficulties in these patients significantly decrease the health-related quality of life. At the same time, fatigue has a negative impact both on the mental and physical components of Health-Related Quality of Life (HRQOL), while cognitive disorders have a greater impact on the mental component of HRQOL [56]. Another study showed that social interactions, cognitive tasks, sleep, recreation, household activities, and hobbies were more impaired in patients with sarcoidosis compared to those in the control group [57].

When it comes to the general quality of life of patients with sarcoidosis, it has been observed that there are several factors, often non-specific, that can influence this [50]. Some researchers have sought to identify the areas of QOL in which patients with sarcoidosis present problems. They pointed out that people suffering from sarcoidosis, both those with or without current symptoms, presented sleep disorders and fatigue that led to a decrease in the quality of life compared to an age-matched control group. In addition to these, the reduction in mobility and in the activities of daily life and the decrease in work capacity have considerably influenced the life of patients with sarcoidosis [57]. In another study, it was shown that QOL in female patients is lower than in male patients, except for the domain of positive feelings [50,58].

On the other hand, the state of health of patients with sarcoidosis was significantly influenced by sleep disorders and specific depressive symptoms [50,57]. It was shown that 60% of patients with sarcoidosis presented depression according to the Depression Scale of the Center for Epidemiological Studies (CES-D) [59]. Also, female gender, systemic sarcoidosis, and the lack of access to medical services are predisposing factors to develop depression in people with sarcoidosis [43].

A study of 64 patients with sarcoidosis showed that 18% presented a Beck Depression Inventory (BDI) score higher than 15 points, which is significant depression. It should be noted that, of the 12 patients, only one had asymptomatic sarcoidosis, which leads to the conclusion that people with symptomatic sarcoidosis have a higher risk of developing depression [57]. Even though there are many studies that show a higher incidence of depression in those suffering from sarcoidosis, its psychiatric implications have not been studied except for the impact on quality of life [43].

There is a lack of data in the medical literature regarding other neurological disorders in patients with sarcoidosis, especially pulmonary fibrosis due to sarcoidosis, and there is a need for more extensive studies on the relationship between the neurological status and pulmonary sarcoidosis.

### 7.3. Neurological Dysfunction in Fibrosing Hypersensitivity Pneumonitis

Until now, there have been no studies focused on neurological disorders in patients with fibrosing hypersensitivity pneumonitis. However, the QOL aspect was studied and compared with other ILDs. Patients with chronic hypersensitivity pneumonia, in addition to the common respiratory symptoms, also have a lower HRQOL compared to ILDs in terms of generic HRQOL measures [60,61].

### 7.4. Neurological Dysfunction in Connective Tissue Diseases

#### 7.4.1. Neurologic Impairment in Systemic Lupus Erythematosus (SLE)

Several neuropsychiatric modifications have been described in SLE patients. The physiopathology of the neurologic impairment includes the rupture of the hematoencephalic barrier, the vascular and neuronal damage due to the abnormal production of cytokines, as well as the action of autoantibodies [62].

Neuropsychiatric SLE is a prevalent manifestation of lupus, affecting 14 to 80% of patients with SLE. It is associated with increased morbidity and mortality [63,64]. The clinical spectrum includes severe symptoms like psychosis, cerebrovascular accident, and myelopathy, as well as chronic ones like headache and cognitive dysfunction. The American College of Rheumatology research committee has established case definitions for 19 neuropsychiatric syndromes involving the central and peripheral nervous systems [65]. Cognitive dysfunction is among them.

Cognitive dysfunction in patients with SLE is among the first neurological manifestations. Difficulties in cognitive abilities were often encountered in patients with SLE with or in the absence of a stroke, a significant psychiatric impairment, or epilepsy [66]. Cognitive impairment has been reported in more than 5% of SLE patients, and up to 3–5% exhibit rapidly progressive or severe forms [67]. A prospective study that included 123 patients with SLE showed that the main predictors of the occurrence of cognitive disorders are diabetes, the persistent identification of antiphospholipid antibodies, depression, the permanent use of prednisone, as well as a low educational level [68].

Studies show varying impairment domains in SLE, including verbal fluency, visual skills, memory, attention, and executive function. Several neuropsychological test batteries have been developed to be sensitive to the types of cognitive deficits associated with neuropsychiatric manifestations of SLE, including the American College of Rheumatology Neuropsychological Test Battery [69] and the Automated Neuropsychological Assessment Metrics (ANAM) Testing [70]. Magnetic resonance imaging (MRI) is commonly used to examine brain lesions in patients with neuropsychiatric SLE, but these lesions may not accurately reflect the disease activity. MRI lesions correlate with cognitive impairment in 72% of SLE patients, as confirmed by neuropsychological testing, with T1- and T2-weighted lesions and cerebral atrophy more common in patients with neuropsychiatric manifestations compared to general population controls [71,72]. Cerebral atrophy is associated with cognitive dysfunction, seizures, and cerebrovascular disease, while T1- and T2-weighted lesions are more specifically associated with seizures and cognitive dysfunction. Other neuroimaging modalities include proton magnetic resonance spectroscopy, functional MRI (fMRI), single photon emission computed tomography (SPECT), and positron emission tomography (PET), with most studies being pilot investigations with small sample sizes.

The impact of cognitive dysfunction in these patients was shown to be on functional outcomes, employment, and health-related quality of life [12,13,14,15,16]. It was also reported that the decrease in cognitive function in patients with SLE was associated with the presence of fatigue, depression, and pain, which can exacerbate the impairment [66]. Furthermore, depression is frequently present in SLE patients with and without overt neuropsychiatric manifestations.

#### 7.4.2. Neurologic Impairment in Rheumatoid Arthritis

Regarding rheumatoid arthritis (RA), it is known that lung damage is present in approximately 40% of patients and is one of the most frequent extra-articular manifestations of the disease [73]. The lungs can be affected in any compartment [74]. However, the extent of neurological damage in patients with rheumatoid arthritis-associated ILD is not known. The vascular ischemia present in RA leads to neuronal demyelination and axonal degeneration. The most common neurologic manifestations are sensory–motor and distal sensory neuropathies, multiple mononeuritis, and other neuropathies, all of which occur in up to 20% of patients [75].

While not as clearly shown as in SLE, RA has been associated with neuropsychiatric symptoms, such as memory, attention, and executive function impairments. Few studies have examined cognitive function in RA; even fewer have included healthy comparison groups or imaging studies. Despite the challenges in cross-study comparisons, some studies found greater cognitive impairment in RA than in healthy controls and less impairment in RA than in SLE. This suggests that disease mechanisms specific to SLE may contribute to the more prevalent cognitive dysfunction in that disorder as compared to RA, an autoimmune disease involving inflammation and pain. Joint pain, stiffness, and RA-related factors may impact cognition function in RA [76].

Bartolini and colleagues conducted a study investigating the impact of CNS alterations in RA on behavior in 30 patients. The mean age of the patients was 55.6 years, with an average disease duration of 11.8 years. Patients with motor impairment due to joint deformities were excluded from the sample, as were those with current depression and previous psychiatric or neurological history. The patients underwent cerebral MRI scans, SPECT, and a 2 h neuropsychological battery, including attention, memory, visual–spatial, and executive function tests. Only two patients performed in the normal range on all tasks. Visuospatial planning ability was impaired in 71% of patients, and visual memory was impaired in 50%. Phonemic verbal fluency was affected in 44%, while only 6% of cases had semantic verbal fluency problems, suggesting more evident left frontal involvement. Verbal memory was impaired in 35% [77].

The study correlated neuropsychological test results with clinical evaluations, including swollen joint count, Ritchie articular index, morning stiffness, erythrocyte sedimentation rate, C-reactive protein, and overall disease severity using the Lee functional index. Impairment on specific tests was not correlated with clinical parameters. However, impairment in Block Design (requiring the manual manipulation of blocks under time constraints) was associated with swollen joint count, the articular index, and Lee functional impairment. Mental flexibility on Trails B and the Wisconsin Card Sorting Test (WCST) was also associated with the Lee scale. In multivariable regression analysis, age, education, disease duration, and disease severity indices were independent variables. On MRI, 35% of patients showed white matter hyperintensities, with low scores on attentional, executive function, and visuospatial tests. On SPECT, hypoperfusion was evident in the frontal lobes in 85% of patients and the parietal lobes in 40%. The authors postulated that motor impairment could be due to microangiopathy in the subcortical and parietal–frontal areas, and joint pain and stiffness could lead to sensory changes affecting motor planning processes [77].

Recent studies have analyzed the possibility of cognitive dysfunction in patients with RA, which is responsible for their QOL [78]. One study found that almost 40% of RA patients have different cognitive domains compromised, such as memory, changes in attention, verbal function, as well as the presence of depressive or anxiety symptoms [75,79].

One study compared three groups of individuals: one with RA, one with sarcoidosis, and one with healthy subjects. The study showed that, although both groups of patients had a poor quality of life, it decreased significantly among patients with RA. The presence of pain and reduced mobility were the main factors contributing to poor quality of life in these patients [50].

#### 7.4.3. Neurologic Impairment in Scleroderma

The neurological implications of scleroderma have been studied for a long time. Research reports show that neurological complications of systemic scleroderma can occur in between 1 and 40% of patients. Initially, it was considered that, in systemic scleroderma, the central nervous system is not usually directly involved, and neurological manifestations are only a consequence of SSc complications of scleroderma. However, in recent years, in addition to the known implications of scleroderma (pulmonary, renal, cardiovascular), more evidence of primary neurological impairment in these patients has begun to appear. Patients who express anti-Scl70 and anti-U1 ribonucleoprotein antibodies (anti-U1RNP) seem to have an increased risk of neurological complications [80].

Cognitive impairment is present in 8.47% of patients with SSc, and it is suggested to be related to the alteration of cerebral perfusion. Moreover, this cognitive dysfunction is more severe in the problem-solving and visual–spatial abilities and tends to progress to dementia [81]. The presence of symptoms of anxiety and depression is very common in patients with scleroderma (73.15% and 23.95%, respectively), highlighting a need for screening in the diagnostic algorithm of this population. Other neurological alterations, also common in these patients, are headaches (23%) and seizures (13.5%) [82]. However, there is no evidence in the medical literature showing that patients with scleroderma-ILD exhibit more significant neurological impairment than those without lung involvement [82].

#### 7.4.4. Neurologic Impairment in Sjogren’s Syndrome

A few studies have analyzed primary Sjogren’s syndrome and found that almost 20% of the patients had neurologic dysfunctions [83]. Cognitive impairment was very common, the prevalence varying across studies from 11% to 100%, depending on the population and methods of assessment [84]. Researchers found that cognitive dysfunction is primarily affected in memory and executive function areas, even without neurological signs or MRI abnormalities. Studies suggest cerebral anti-muscarinic acetylcholine receptor (mAChR) autoantibodies may play a pathogenic role in immune-mediated neuroinflammation and cognitive dysfunction in patients with Sjogren’s syndrome frontal lobe syndrome (slowness, shifting capacity disorder, incapacity to resist cognitive conflict, programming capacity disorder, and decreased verbal fluency) [85]. The patients present attention, memory, and executive disorders, with difficulty concentrating, especially when following a discussion, finding their words, forgetting what they are looking for, or not finishing what they have started [86]. Indart et al. found that the WCST was the most sensitive test for analyzing the neuropsychological profile of these subjects. They found that up to 80% had cognitive impairment, especially in the executive domain; they could not keep a complex thought in memory or find adaptive strategies to a particular situation due to impaired capacity of abstraction or difficulties adapting to external reactions [87].

Further, their central nervous system (CNS) involvement is supported by cranial MRI findings, which show multifocal lesions similar to those encountered in multiple sclerosis. Other common neurological manifestations in patients with Sjogren’s syndrome include CNS involvement (e.g., transverse myelitis), cranial neuropathies (e.g., optic neuritis), myopathy, painful small-fiber neuropathy, as well as peripheral neuropathies [84].

It should be mentioned that, in the case of patients with Sjogren’s syndrome, as in other types of connective tissue diseases with possible pulmonary involvement, it has not been shown until now if those with impaired lung function have more significant neurological implications than those with normal respiratory function.

#### 7.4.5. Neurologic Impairment in Polymyositis and Dermatomyositis

Until now, no studies have analyzed the possible neurological involvement in polymyositis and dermatomyositis with pulmonary determination.

### 7.5. Neurological Dysfunction in Vasculitis

In patients with vasculitis, it has been reported that there is a two-fold higher risk of experiencing severe fatigue compared to the general population [88]. Moreover, this severe fatigue decreases the quality of life by 75% for these patients [89,90,91].

Moreover, one study highlighted the association between perseverative cognition and fatigue in patients with vasculitis because it is considered that the stress–fatigue link has perseverative cognition as a key factor. In other words, the cognitive demand in these patients is more significant, as an additional effort is required to continue carrying out some tasks that can lead to behavioral disengagement or higher fatigue [91].

Lung involvement is most commonly observed with primary idiopathic vasculitis of small vessels: granulomatosis with polyangiitis, microscopic polyangiitis, Churg–Strauss syndrome, idiopathic pauci-immune rapidly progressive glomerulonephritis, and isolated pauci-immune pulmonary capillaritis [92].

When it comes to granulomatosis with polyangiitis, several manifestations in the central and peripheral nervous system are described in the literature. Among these are headaches, cerebrovascular accidents, convulsions, meningitis, spinal cord injuries, motor and sensory peripheral neuropathies, cranial nerve paralysis, brain mass injuries, and sensorineural hearing loss [93].

In Churg–Strauss syndrome, peripheral neurological symptoms are present more often than those of the CNS [84].

However, none of these studies focused on the relationship between the severity of the lung involvement and the degree of neurologic impairment.

### 7.6. Neurologic Impairment in NSIP and Unclassifiable ILDs

Regarding NSIP and unclassifiable ILDs, there are very little data in the literature associating them with neurological damage. One study included 51 patients with ILD, of which 12 had unspecified ILD and the rest had various other types of ILD (IPF, sarcoidosis, pneumonia in cryptogenic organization, pneumonitis, unspecified interstitial pneumonia, pulmonary histiocytosis). The conclusions of the study showed that these patients had impaired cognition at the levels of visual and verbal memory, working memory, and visual perception, but processing speed, attention, and executive functions were intact. At the same time, the authors suggested that certain cardiopulmonary measurements are predictors for neuropsychic functionality in patients with ILD, including those with unclassified ones. Heart rate after effort, oxygen saturation before and after exercise, and the distance covered in the walking test are good predictors for psychomotor speed, verbal fluency, and verbal memory measurement. The diffusion capacity of the lungs for carbon monoxide correlates with verbal learning, fluency in expression, and visual memory. These measurements cannot predict attention span, visual perception, or memory [8].

On the other hand, the frequency of comorbidities in unclassifiable ILDs is lower than in other types of ILDs. However, the most common comorbidities reported by a study are depression and cerebrovascular diseases. The impact of comorbidities on mortality is not yet known [94].

Table 1 summarizes the most important evidence regarding neurologic involvement, cognitive impairment, anxiety, depression, and quality of life in the most common fibrosing interstitial lung diseases.

## 8. Discussion

Progressive fibrosing ILDs are generally considered rare conditions, characterized by intricate pathophysiology, diverse clinical presentations, and unfavorable outcomes. This review emphasizes the current understanding of neurological cognitive function and the prevalence of depression and anxiety among patients with particular types of fibrosing ILDs.

The studies that we analyzed showed a large variability in the screening methods used for neuropsychological impairment detection, diagnostic criteria, and terminology. The implementation of standard screening methods, terminology, and classification for neurological dysfunction in this group of individuals could lead to obtaining more accurate data on the real prevalence of these deficiencies.

Neurological manifestations associated with ILDs can arise at any point during the progression of the illness and significantly influence the overall prognosis, although the underlying pathophysiological mechanisms remain largely undefined.

Cognition represents the most extensively researched neurological function, regardless of the ILD subtype, and impairments can interfere at any of its domains. This long-lasting impairment may be permanent, but more research is needed to understand its prevalence, nature, risk factors, etiology, and nuances.

We also noticed that, currently, there is a complete lack of data concerning other neurological disorders, besides cognitive function, anxiety, and depression, in patients with fibrosing ILDs. Consequently, additional studies are essential to enhance our understanding of the relationship between ILDs and neuropsychiatric dysfunction.

In non-critical care clinical settings, many physicians fail to recognize or assess neurocognitive impairment in 35–90% of patients [95]. It is important to identify them because these conditions can result in functional disabilities, adverse clinical outcomes, and reduced adherence to treatment, ultimately contributing to a diminished quality of life and increased social isolation.

In addition, even though anxiety and depression symptoms are present in 35% and 37% of IPF patients, respectively, less than 5% were prescribed medication for mood disorders [96], indicating that they are underrecognized and undertreated. Even though antifibrotic medication prescribed for patients with IPF and for progressive ILDs is not an antipsychotic treatment, by improving dyspnea, cough, and exercise capacity, it can reduce the incidence and severity of anxiety and depression [97].

Moreover, there is also a complete lack of data regarding neurologic rehabilitation programs for patients with ILDs. Several studies following the benefits of pulmonary rehabilitation showed a significant reduction in symptoms of anxiety and depression in the ILD population [96,98]. Future research should concentrate on the immediate determinants of both neurological impairment and mood disorders and also on the most feasible interventions aimed to prevent neurocognitive morbidity. They may provide significant insights into the detection, natural progression, prognosis, and underlying mechanisms of neurophysiological deficits. This will guide the development, implementation, and fine-tuning of intervention programs.

Increased identification of neurocognitive impairments may benefit patients by raising physician awareness and potentially leading to increased referrals to rehabilitation specialists, neuropsychologists, speech and language therapists, and other healthcare providers who can provide interventions such as cognitive remediation.

## 9. Conclusions

Most progressive-fibrosing ILDs are rare diseases with complex pathogeny, heterogenic clinical manifestations, and poor prognosis. In general, neurocognitive and neuropsychiatric sequelae are common in patients with respiratory disorders, particularly those with concomitant hypoxia. Neurological manifestations of ILDs can emerge at any stage of the disease, and they have an important impact on the prognosis. An approach focused on the early diagnosis of cognitive impairment should be included in the management algorithm of interstitial lung diseases, considering the consequences of these manifestations on the quality of life and morbidity of this disease. Research should focus on proximal determinants and interventions to prevent neurocognitive morbidity, yielding valuable insights into the identification, natural history, prognosis, and potential mechanisms of neurocognitive deficits.

## Figures and Tables

**Figure 1 biomedicines-12-02572-f001:**
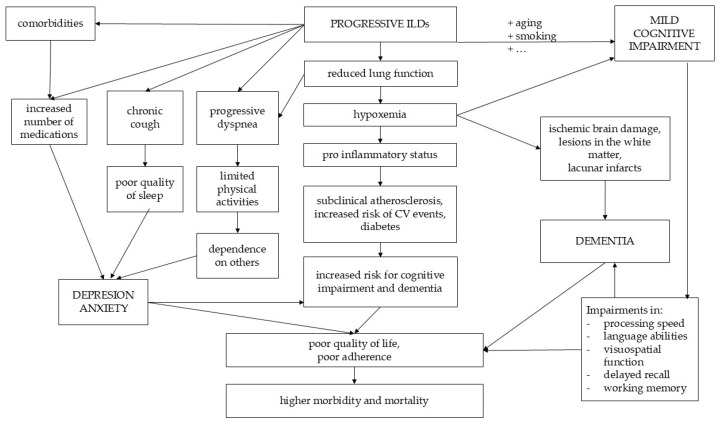
The pathogenesis of neurological damage in fibrosing ILD.

**Table 1 biomedicines-12-02572-t001:** Neurocognitive and neuropsychiatric implications of fibrosing interstitial lung diseases.

	Neurologic Involvement	Cognitive Impairment	Cognitive Impairment Pathogenesis	Anxiety	Depression	Quality of Life
**Idiopathic pulmonary fibrosis (IPF)**	Prevalence: unknown; Cognitive impairment	Most affected domains: understanding, recalling, working memory, language, and visuospatial skills [10,32]	Aging, smoking status, hypoxemia, anxiety, and depression [10].	up to 60%, [44,45,46]	24.3–49.2% [44,45,46]	↓ due to lack of independence and poor physical health
**Sarcoidosis**	Prevalence: unknown; Most common: cognitive impairment, hypothalamus, cranial nerves, and pituitary gland evolvement [33,43,55].	Prevalence: up to 33% [8,33,53]	Small-fiber neuropathy and fatigue [52], disease duration [33].	up to 60% [59]	+ [59]	↓ due to sleep disorders, fatigue, reduction in mobility, in the activities of daily life, in working capacity, depression [43,57]
**Systemic lupus erythematosus**	Prevalence: 14 to 80% of patients [63,64]; Most common: cognitive dysfunction, headache, cerebrovascular accident, psychosis, epilepsy, myelopathy [65,71,72].	Prevalence: approximately 5%. Most affected domains: verbal fluency, visual skills, memory, attention, and executive function [69,70].	Rupture of the hematoencephalic barrier, the vascular and neuronal damage due to the abnormal production of cytokines and autoantibodies [62].	+ [66]	+ [66]	↓ [12,13,14,15,16]
**Rheumatoid arthritis**	Prevalence: up to 20% of patients; Most common: cognitive impairment, sensory–motor and distal sensory neuropathies, multiple mononeuritis, and other neuropathies [75,76].	Prevalence: up to 40%. Most affected domains: memory, attention, executive function, visuospatial planning, verbal fluency [77].	Vascular ischemia leads to neuronal demyelination and axonal degeneration [77].	+ [75,79]	+ [75,79]	↓↓ [50,78]
**Scleroderma**	Prevalence: 1–40% of patients [80]; Most common: cognitive impairment, headaches, and seizures [82].	Prevalence: 8.47%. Most affected domains: problem-solving, visual–spatial abilities Tends to progress to dementia [81].	Positive anti-Scl70 and anti-U1 ribonucleoprotein antibodies, alteration of cerebral perfusion [80].	up to 73.15% [82]	up to 23.95% [82]	-
**Sjogren’s syndrome**	Prevalence: up to 20% of patients [83]; Most common: cognitive impairment, transverse myelitis, cranial neuropathies, myopathy, painful small-fiber neuropathy, peripheral neuropathies [84].	Prevalence: 11–100% [84]. Most affected domains: memory, executive function, slowness, shifting capacity disorder, incapacity to resist cognitive conflict, programming capacity disorder, verbal fluency, attention, difficulty concentrating [85,86,87].	Anti-muscarinic acetylcholine receptor (mAChR) autoantibodies, neuroinflammation [85].	-	-	
**Vasculitis**	Prevalence: unknown; Most common: headaches, cerebrovascular accidents, convulsions, meningitis, spinal cord injuries, motor and sensory peripheral neuropathies, cranial nerve paralysis, brain mass injuries, sensorineural hearing loss [84,93].		Stress–fatigue link [91].	-	-	Severe fatigue decreases [89,90,91]
**Nonspecific interstitial pneumonia (NSIP) and unclassifiable ILDs**	Prevalence: unknown	Most affected domains: visual and verbal memory, working memory, and visual perception [8]	Tachycardia, hypoxemia, reduced effort capacity [8]	+ [94]	+ [94]	
**Hypersensitivity pneumonitis, dermatomyositis, polymyositis**	Prevalence: unknown	-		-	-	↓ [60,61]

## Data Availability

The data are encapsulated within this article. Further details can be obtained upon request from either the primary author or the corresponding author.

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
