# Peer review of "Neurocognitive and Neuropsychiatric Implications of Fibrosing Interstitial Lung Diseases"

_biomedicines, 2024, doi:10.3390/biomedicines12112572_

Round 1

Reviewer 1 Report

Comments and Suggestions for Authors

General comments:

Very interesting and novel project with few published manuscripts on the subject. However, some minor comments are made in order to improve the current versión.

Perhaps it would have been more interesting for the reader not only to do a narrative review, but also a review, although not systematic, providing tables that summarize the evidence available in each of the sections that are reviewed, epidemiology, diagnosis, etc.

There is a lack of a better explanation of the methodology, terms used, study time, number of studies included, etc. Likewise, the discussion section is missing.

The conclusions are extensive. Rate summarize.Maybe “Increased identification 589 of neurocognitive impairments may benefit patients by raising physician awareness…” could be included in the final part of the discussion section

Specific comments:

.- Abstract. Perhaps it is not necessary or pertinent in this section to refer to all the interstitial lung diseases that were included in the review. As is well known in the material and methods section, all the pathologies that are included in the review must be specified.

Methodology and discussion are missing.

.- Keywords. 9 terms are included. Maybe they are so much. Usually, 5 or 6 terms are included. Please, review.

.- Line 266. “On the other hand” and in line 283 the same phrase. Line 362 and 541, idem. Please, review.

 .- 6.1. Neurological dysfunction in IPF. This section refers to comorbidities, cognitive impairment, depression and anxiety. Perhaps it would be worth subdividing these pathologies in relation to IPF.

.- References. 95 quotes are included of which 24 (25.26%) are recent, this is five years or less from its publication. I would be interested in considering including any additional quotes.

Some quotes appear incomplete, such as quotes 11, 35, 50 and 70. Please, review.

In some references the full name of the journal or its name appears in capital letters. Please, review.

Author Response

Thank you very much for taking the time to review this manuscript. We find that your comments and suggestions added great value to our narrative review. We really appreciate! Please find the detailed responses below and the corresponding revisions in highlights in the re-submitted files.

Comments 1. Perhaps it would have been more interesting for the reader not only to do a narrative review, but also a review, although not systematic, providing tables that summarize the evidence available in each of the sections that are reviewed, epidemiology, diagnosis, etc.

Response 1: Thank you for your suggestion. We introduced a table which summarizes the most important evidence regarding neurologic involvement, cognitive impairment, anxiety, depression and quality of life, in the most common fibrosing interstitial lung diseases. Please find the table at page 15.

Comments 2. There is a lack of a better explanation of the methodology, terms used, study time, number of studies included, etc. Likewise, the discussion section is missing.

Response 2: Thank you. In the revised manuscript, we added more information on the methodology used in our review (page 3, line 91-103) [We performed a comprehensive literature search in the PubMed database to identify relevant studies describing the impact of ILDs on several neuropsychiatric domains from inception to march 2024. A combination of search terms "interstitial lung disease," "pulmonary fibrosis", “IPF”, “sarcoidosis”, “hypersensitivity pneumonitis”, “connective tissue diseases”, “systemic lupus erythematous” , “rheumatoid arthritis”, “scleroderma”, “Sjogren syndrome”, “polymyositis”, “dermatomyositis”, “vasculitis”, “nonspecific interstitial pneumonia”, “unclassifiable ILDs”, “neurologic”, “cognitive”,” central nervous system”, “anxiety”, “depression” were used to identify English language studies, that had the key search terms in their title or abstract. In our review, we included original research on neurological and psychiatric manifestations in adult subjects with ILDs. The scoping search identified 5153 articles, of which 74 were directly related to our research topic.]. Nonetheless, we performed a scoping search (a brief literature search to understand the extent of the research on that topic), not a systematic review. We also added a discussion section (page 18, line 608-653) [Progressive fibrosing ILDs are generally considered rare conditions, characterized by intricate pathophysiology, diverse clinical presentations, and unfavorable outcomes. This review emphasizes the current understanding of neurological function cognitive and the prevalence of depression and anxiety among patients with particular types of fibrosing ILDs.

Between the studies that we analyzed, we found a large variability in the screening methods used for neuro-phycological impairment detection, diagnostic criteria and terminology. The implementation of standard screening methods, terminology and classification for neurological dysfunction in this group of individuals, could lead to obtaining more accurate data on the real prevalence of these deficiencies.

Neurological manifestations associated with ILDs can arise at any point during the progression of the illness and significantly influence the overall prognosis, although the underlying pathophysiological mechanisms remain largely undefined.

Cognition represents the most extensively researched neurological function, regardless of the ILD subtype, and impairments can interfere at any of its domains. This long-lasting impairment may be permanent, but more research is needed to understand it’s prevalence, nature, risk factors, etiology, and nuances.

We also noticed that currently there is a complete lack of data concerning other neurological disorders, besides cognitive function, anxiety and depression, in patients with fibrosing ILDs. Consequently, additional studies are essential to enhance our understanding of the relationship between ILDs and neuropsychiatric dysfunction.

In non-critical care clinical settings, many physicians fail to recognize or assess neurocognitive impairment in 35-90% of patients [95]. It is important to identify them be-cause these conditions can result in functional disabilities, adverse clinical outcomes, and reduced adherence to treatment, ultimately contributing to a diminished quality of life and increased social isolation.

Besides, even though anxiety and depression symptoms are present in 35% respectively 37% of IPF patients, less than 5% were prescribed medication for mood disorders [96], indicating they are underrecognized and undertreated. Even tough antifibrotic medication prescribed for patients with IPF and for progressive ILDs, is not an antipsychotic treatment, by improving dyspnea, cough and exercise capacity, can reduce the incidence and severity of anxiety and depression [97].

Moreover, there is also a complete lack of data regarding neurologic rehabilitation programs for patients with ILDs. Several studies following the benefits of pulmonary rehabilitation showed a significant reduction in symptoms of anxiety and depression in ILD population [96,98]. Future research should concentrate on the immediate determinants of both neurological impairment and mood disorders, and also on the most feasible interventions aimed to prevent neurocognitive morbidity. They may provide significant insights into the detection, natural progression, prognosis, and underlying mechanisms of neuro-physiological deficits. This will guide the development, implementation, and fine-tuning of intervention programs.

Increased identification of neurocognitive impairments may benefit patients by raising physician awareness and potentially leading to increased referrals to rehabilitation specialists, neuropsychologists, speech and language therapists, and other healthcare providers who can provide interventions such as cognitive remediation.]

Comments 3. The conclusions are extensiveRate summarize. Maybe “Increased identification 589 of neurocognitive impairments may benefit patients by raising physician awareness…” could be included in the final part of the discussion section

Response 3: Thank you for your suggestion. We revised the conclusion section and made it more concise and introduced the phrase Increased identification of neurocognitive impairments may benefit patients by raising physician awareness…” in the discussion section (page 18, line 650)

Specific comments:

Comments 4. Abstract. Perhaps it is not necessary or pertinent in this section to refer to all the interstitial lung diseases that were included in the review. As is well known in the material and methods section, all the pathologies that are included in the review must be specified.

Response 4: Thank you for pointing this out. Therefore, we have revised the abstract and made it more concise (page 1, line 21-30). [Abstract: Patients with interstitial lung diseases (ILDs) associate a large variety of comorbidities that have a significant impact on their clinical outcomes and survival. Among these comorbidities there is neurological impairment. This review highlights what is known about the cognitive function, central nervous system (CNS), depression and anxiety in patients with specific forms of fibrosing ILDs such as idiopathic pulmonary fibrosis, sarcoidosis, hypersensitivity pneumonitis, connective tissue diseases, etc. The most common pathogenic mechanisms for neurocognitive dysfunction, as well as the screening methods and tools for their identification, will also be de-scribed in this review.]

Comments 5. Methodology and discussion are missing.

Response 5: Thank you for your suggestion. We introduced methodology (page 3, line 91-103)

 and discussion sections (page 17, line 608-653).

Comments 6. Keywords. 9 terms are included. Maybe they are so much. Usually, 5 or 6 terms are included. Please, review.

Response 6: Thank you for pointing this out. We revised the keywords and reduced them to 6 terms (page 1, line 31-33) [Keywords: cognitive impairment; depression; anxiety; idiopathic pulmonary fibrosis; sarcoidosis; connective tissue diseases;]

Comments 7. Line 266. “On the other hand” and in line 283 the same phrase. Line 362 and 541, idem. Please, review.

Response 7: Thank you for your suggestion. We replaced the terms. (page 7, line 303 and page 15, line 563)

Comments 8. Neurological dysfunction in IPF. This section refers to comorbidities, cognitive impairment, depression and anxiety. Perhaps it would be worth subdividing these pathologies in relation to IPF.

Response 8: Thank you for pointing this out. We have subdivided these comorbidities.

7.1.1 Cognitive impairment in IPF (page 6, line 266)

7.1.2 Anxiety and depression in IPF (page 7, line 291)

7.1.3 Quality of life (QOL) in IPF (page 7, line 317)

Comments 9. References. 95 quotes are included of which 24 (25.26%) are recent, this is five years or less from its publication. I would be interested in considering including any additional quotes.

Some quotes appear incomplete, such as quotes 11, 35, 50 and 70. Please, review.

In some references the full name of the journal or its name appears in capital letters. Please, review.

Response 9 Due to the lack of data, except maybe regarding IPF, we included all the studies we found and unfortunately few of them are recent. Nonetheless, we introduced 3 more recent articles [96.Edwards GD, Polgar O, Patel S, Barker RE, Walsh JA, Harvey J, et al. Mood disorder in idiopathic pulmonary fibrosis: response to pulmonary rehabilitation. ERJ Open Res. 2023 May 22;9(3):00585-2022.

97.He X, Ji J, Pei Z, Luo Z, Fang S, Liu X, et al. Anxiety and depression status in patients with idiopathic pulmonary fibrosis and outcomes of nintedanib treatment: an observational study. Ann Med. 2024 Dec;56(1):2323097.

98.Deniz S, Åžahin H, Yalnız E. Does the severity of interstitial lung disease affect the gains from pulmonary rehabilitation? Clin Respir J 2018; 12: 2141–2150.]. 

Thank you also for pointing out the quotes who were incomplete. The reference manger program missed adding some data or introduced them in capital letters. Our fault that we didn’t properly check all of them. We revised the bibliography as you requested. (page 19-23)

Reviewer 2 Report

Comments and Suggestions for Authors

The manuscript entitled: Neurological Implications of Fibrosing Interstitial Lung Diseases, highlights what is known about the cognitive function, central nervous system (CNS), depression and anxiety in patients with specific forms of fibrosing ILDs. It is interesting and well written, however some issues need to be revised and edited. Please, find the detailed comments below:

1         I think the title need to be changed to neuropsychiatric implications sine the authors are discussing depression, anxiety and quality of life impairment.

2         In the following paragraph entitled: 3. Diagnosis and classification of interstitial lung diseases. From line 123 o 124 the following sentence is written “This section may be divided by subheadings. It should provide a concise and precise description of the experimental results, their interpretation, as well as the experimental conclusions that can be drawn. What are these comments come from? These sentences need to be omitted.

3         In line 177: This review will discuss in more detail the particular aspects of neurological damage in the different subtypes of fibrosing ILD. Need to be omitted. The authors already said that many times before.

4         In line 163: the authors referred to Elfferich et al. study as reference 33, which is mismatched with the references list at the end of the manuscript.

5         I suggest that the diagnosis and classification, or the pathogenesis summarized as a  diagram for more informative representation of the author’s ideas.  

6         This paragraph from line 212-219 “An important characteristic of patients with fibrosing ILD, especially those with IPF, is 212 progressive dyspnea that leads to a marked limitation of physical activities and, over time, 213 increases their dependence on those around them, even for simple tasks. Due to these 214 lifestyle changes, many of the patients develop symptoms such as anxiety and depression, 215 which significantly reduces their quality of life [41]. Depression is also associated with 216 certain variables, such as poor quality of sleep or chronic pain. Taking into consideration 217 ILD features and depression contributing factors, patients must also receive an antidepressant treatment, as it is unlikely to resolve this without it [34].”, is better placed in line 196 for more convenient flow of ideas.

7         In line 274 the authors referred to De Vries et al. study as reference number [43], which is mismatched with the references list at the end of the manuscript.

8         In line 283: The authors stated this fact “On the other hand, depression has a significant role in reducing the quality of life of 283 subjects with IPF [46]” several times throughout the manuscript. I just can notice I have already read this many times. Please, revise this and make sure that every paragraph discusses a new idea.

9         The paragraph from line 291 the authors stated that “Another study on patients with IPF investigated the major issues that decrease their quality of life through discussion groups. Among the significant aspects of a good quality of life were mobility, hobbies or other leisure activities, transport, work capacity, energy, and social relations. Interestingly, these patients preferred the World Health Organization Quality-of-Life Scale (WHOQOL-100) questionnaire to assess quality of life rather than the classic SGRQ [48, 49]”. It doesn’t add important knowledge here, it would be better to be omitted.

10     In line 297 the authors stated that “Quality of life (QOL) The relation of depressive symptoms and dyspnea with the QOL in IPF patients was also investigated.” This sentence needs to be revised is Quality of life (QOL) as subtitle? Also it has been discussed in details in the previous paragraphs.

11     In line 312 the authors stated that “Regarding other neurological disorders”. What are the other neurological disorders you mean?

12     Again in line 375 “regarding other neurological disorders” What are the other neurological disorders the authors mean?

13     It would be better to add a summarizing table for the studies analyzed the neurological implications of the most common types of fibrosing ILD including IPF, sarcoidosis, connective tissue diseases, ….. etc. also, please demonstrate the main characteristics of these studies including number of patients, location, year, study design, and main findings.

14     One of the major limitations of this review is only using the PubMed database for searching without including other databases for more comprehensive review such as Google scholar, OVID, or Cochrane.

15     Although the conclusion is adequately written and comprehensive, it raised a concern about interventions. Have the authors found any studies that discuss the interventions applied for the rehabilitation of the neurological impairments, if so, why not discussed even briefly at the end just before the conclusion section.

Author Response

Thank you very much for taking the time to review this manuscript. We find that your comments and suggestions added great value to our narrative review. We really appreciate! Please find the detailed responses below and the corresponding revisions in highlights in the re-submitted files.

Comments 1. I think the title need to be changed to neuropsychiatric implications sine the authors are discussing depression, anxiety and quality of life impairment.

Response 1: Thank you for your suggestion. We changed our title in “Neurocognitive and neuropsychiatric implications of Fibrosing Interstitial Lung Diseases

Comments 2.   In the following paragraph entitled: 3. Diagnosis and classification of interstitial lung diseases. From line 123 o 124 the following sentence is written “This section may be divided by subheadings. It should provide a concise and precise description of the experimental results, their interpretation, as well as the experimental conclusions that can be drawn. What are these comments come from? These sentences need to be omitted.

Response 2: Thank you for your comment. It was from the initial draft and we omitted to delete it (Page 3, Line 128)

Comments 3.   In line 177: This review will discuss in more detail the particular aspects of neurological damage in the different subtypes of fibrosing ILD. Need to be omitted. The authors already said that many times before.

Response 3: Thank you for pointing this out. We deleted the sentence. (Page 4, Line 182)

Comments 4.   In line 163: the authors referred to Elfferich et al. study as reference 33, which is mismatched with the references list at the end of the manuscript.

Response 4: Thank you for pointing this out. We reviewed the references list. (Page 20, Line 749)

Comments 5.  I suggest that the diagnosis and classification, or the pathogenesis summarized as a diagram for more informative representation of the author’s ideas.  

Response 5: Thank you for your suggestion. We introduced a diagram which summarizes the pathogenesis. (Page 6, Line 247)

Comments 6.  This paragraph from line 212-219 “An important characteristic of patients with fibrosing ILD, especially those with IPF, is 212 progressive dyspnea that leads to a marked limitation of physical activities and, over time, 213 increases their dependence on those around them, even for simple tasks. Due to these 214 lifestyle changes, many of the patients develop symptoms such as anxiety and depression, 215 which significantly reduces their quality of life [41]. Depression is also associated with 216 certain variables, such as poor quality of sleep or chronic pain. Taking into consideration 217 ILD features and depression contributing factors, patients must also receive an antidepressant treatment, as it is unlikely to resolve this without it [34].”, is better placed in line 196 for more convenient flow of ideas.

Response 6: Thank you for your suggestion. We placed the paragraph where you suggested. (Page 5, Line 193)

Comments 7.  In line 274 the authors referred to De Vries et al. study as reference number [43], which is mismatched with the references list at the end of the manuscript.

Response 7: Thank you for pointing this out. We reviewed the references list. (Page 21, Line 769)

Comments 8.  In line 283: The authors stated this fact “On the other hand, depression has a significant role in reducing the quality of life of 283 subjects with IPF [46]” several times throughout the manuscript. I just can notice I have already read this many times. Please, revise this and make sure that every paragraph discusses a new idea.

Response 8: Thank you. We deleted the sentence. (Page 7, Line 303)

Comments 9.  The paragraph from line 291 the authors stated that “Another study on patients with IPF investigated the major issues that decrease their quality of life through discussion groups. Among the significant aspects of a good quality of life were mobility, hobbies or other leisure activities, transport, work capacity, energy, and social relations. Interestingly, these patients preferred the World Health Organization Quality-of-Life Scale (WHOQOL-100) questionnaire to assess quality of life rather than the classic SGRQ [48, 49]”. It doesn’t add important knowledge here, it would be better to be omitted.

Response 9: Thank you for pointing this out. We deleted the paragraph. (Page 7, Line 311)

Comments 10. In line 297 the authors stated that “Quality of life (QOL) The relation of depressive symptoms and dyspnea with the QOL in IPF patients was also investigated.” This sentence needs to be revised is Quality of life (QOL) as subtitle? Also it has been discussed in details in the previous paragraphs.

Response 10: Thank you for pointing this out. Yes, it was supposed to be a different subtitle. We also deleted the phrase discussed before in order not to repeat the same idea, as a response to comment 8. (Page 7 Line 317)

Comments 11.  In line 312 the authors stated that “Regarding other neurological disorders”. What are the other neurological disorders you mean?

Response 11. Thank you for your suggestion. We became more explicit and introduced in more details (nonpsychotic mental disorders, extrapyramidal and movement disorders, epilepsy, nerve, nerve root and plexus disorders, etc) (Page 8, Line 332)

Comments 12. Again in line 375 “regarding other neurological disorders” What are the other neurological disorders the authors mean?

Response 12. Thank you for your suggestion. We became more explicit and introduced more details in page 8, line 332.(nonpsychotic mental disorders, extrapyramidal and movement disorders, epilepsy, nerve, nerve root and plexus disorders, etc)

Comments 13.  It would be better to add a summarizing table for the studies analyzed the neurological implications of the most common types of fibrosing ILD including IPF, sarcoidosis, connective tissue diseases, ….. etc. also, please demonstrate the main characteristics of these studies including number of patients, location, year, study design, and main findings.

Response 13. Although your suggestion is interesting, this is a narrative review and such a detailed characterization is beyond our purpose.

Comments 14.  One of the major limitations of this review is only using the PubMed database for searching without including other databases for more comprehensive review such as Google scholar, OVID, or Cochrane.

Response 14. Thank you for pointing this out. Yes, this is a limitation; however, we do not believe extending our search in other databases would have brought up important and new information.

Comments 15.  Although the conclusion is adequately written and comprehensive, it raised a concern about interventions. Have the authors found any studies that discuss the interventions applied for the rehabilitation of the neurological impairments, if so, why not discussed even briefly at the end just before the conclusion section.

Response 15 Thank you for your suggestion. We changed our conclusion and introduced a discussion section where we briefly presented the interventions and rehabilitation methods. However, there are no data in the literature regarding a specific neurologic rehabilitation in ILDs. Please see the discussion section. (Page 17-18, Line 608-654)

[Progressive fibrosing ILDs are generally considered rare conditions, characterized by intricate pathophysiology, diverse clinical presentations, and unfavorable outcomes. This review emphasizes the current understanding of neurological function cognitive and the prevalence of depression and anxiety among patients with particular types of fibrosing ILDs.

Between the studies that we analyzed, we found a large variability in the screening methods used for neuro-phycological impairment detection, diagnostic criteria and terminology. The implementation of standard screening methods, terminology and classification for neurological dysfunction in this group of individuals, could lead to obtaining more accurate data on the real prevalence of these deficiencies.

Neurological manifestations associated with ILDs can arise at any point during the progression of the illness and significantly influence the overall prognosis, although the underlying pathophysiological mechanisms remain largely undefined.

Cognition represents the most extensively researched neurological function, regardless of the ILD subtype, and impairments can interfere at any of its domains. This long-lasting impairment may be permanent, but more research is needed to understand it’s prevalence, nature, risk factors, etiology, and nuances.

We also noticed that currently there is a complete lack of data concerning other neurological disorders, besides cognitive function, anxiety and depression, in patients with fibrosing ILDs. Consequently, additional studies are essential to enhance our understanding of the relationship between ILDs and neuropsychiatric dysfunction.

In non-critical care clinical settings, many physicians fail to recognize or assess neu-rocognitive impairment in 35-90% of patients [95]. It is important to identify them be-cause these conditions can result in functional disabilities, adverse clinical outcomes, and reduced adherence to treatment, ultimately contributing to a diminished quality of life and increased social isolation.

Besides, even though anxiety and depression symptoms are present in 35% respectively 37% of IPF patients, less than 5% were prescribed medication for mood disorders [96], indicating they are underrecognized and undertreated. Even tough antifibrotic medi-cation prescribed for patients with IPF and for progressive ILDs, is not an antipsychotic treatment, by improving dyspnea, cough and exercise capacity, can reduce the incidence and severity of anxiety and depression [97].

Moreover, there is also a complete lack of data regarding neurologic rehabilitation programs for patients with ILDs. Several studies following the benefits of pulmonary rehabilitation showed a significant reduction in symptoms of anxiety and depression in ILD population [96,98]. Future research should concentrate on the immediate deter-minants of both neurological impairment and mood disorders, and also on the most feasible interventions aimed to prevent neurocognitive morbidity. They may provide significant insights into the detection, natural progression, prognosis, and underlying mechanisms of neuro-physiological deficits. This will guide the development, imple-mentation, and fine-tuning of intervention programs.

Increased identification of neurocognitive impairments may benefit patients by raising physician awareness and potentially leading to increased referrals to rehabilitation specialists, neuropsychologists, speech and language therapists, and other healthcare providers who can provide interventions such as cognitive remediation.]

Reviewer 3 Report

Comments and Suggestions for Authors

Dear authors,

thank you for the opportunity to read and review the manuscript.

It is very interesting and well written.

Fibrosing interstitial lung diseases have an overwhelming impact on neuropsychological and neuropsychiatric function of the patients; keeping in mind the different diseases, their pathophysiology, neurological and psychological impact, early recognition, treatment and rehabilitation together with phycological and psychiatric support could improve quality of life of these patients. 

[-He X, Ji J, Pei Z, Luo Z, Fang S, Liu X, Lei Y, Yan H, Guo L. Anxiety and depression status in patients with idiopathic pulmonary fibrosis and outcomes of nintedanib treatment: an observational study. Ann Med. 2024 Dec;56(1):2323097. doi: 10.1080/07853890.2024.2323097. Epub 2024 Apr 6. PMID: 38581666; PMCID: PMC11000612.

-Delameillieure A, Dobbels F, Fieuws S, Leceuvre K, Vanderauwera S, Wuyts WA. Behavioural and psychological patterns of patients with idiopathic pulmonary fibrosis: a prospective study. Respir Res. 2022 May 14;23(1):124. doi: 10.1186/s12931-022-02041-6. PMID: 35568881; PMCID: PMC9107011.

-Edwards GD, Polgar O, Patel S, Barker RE, Walsh JA, Harvey J, Man WD, Nolan CM. Mood disorder in idiopathic pulmonary fibrosis: response to pulmonary rehabilitation. ERJ Open Res. 2023 May 22;9(3):00585-2022. doi: 10.1183/23120541.00585-2022. PMID: 37228278; PMCID: PMC10204825.]

I found the paper interesting and well written; I suggest modifying the title as the paper does not focus properly and only on neurological implication of fibrosing interstitial lung disease but mostly on neurocognitive and neuropsychiatric impairment. 

Further studies are needed to proper specify for each disease the most appropriate treatment and support required.

Author Response

Thank you very much for taking the time to review this manuscript. We find that your comments and suggestions added great value to our narrative review. We really appreciate! Please find the detailed responses below and the corresponding revisions in highlights in the re-submitted files. 

Comments 1.  Fibrosing interstitial lung diseases have an overwhelming impact on neuropsychological and neuropsychiatric function of the patients; keeping in mind the different diseases, their pathophysiology, neurological and psychological impact, early recognition, treatment and rehabilitation together with phycological and psychiatric support could improve quality of life of these patients. 

[-He X, Ji J, Pei Z, Luo Z, Fang S, Liu X, Lei Y, Yan H, Guo L. Anxiety and depression status in patients with idiopathic pulmonary fibrosis and outcomes of nintedanib treatment: an observational study. Ann Med. 2024 Dec;56(1):2323097. doi: 10.1080/07853890.2024.2323097. Epub 2024 Apr 6. PMID: 38581666; PMCID: PMC11000612.

-Delameillieure A, Dobbels F, Fieuws S, Leceuvre K, Vanderauwera S, Wuyts WA. Behavioural and psychological patterns of patients with idiopathic pulmonary fibrosis: a prospective study. Respir Res. 2022 May 14;23(1):124. doi: 10.1186/s12931-022-02041-6. PMID: 35568881; PMCID: PMC9107011.

-Edwards GD, Polgar O, Patel S, Barker RE, Walsh JA, Harvey J, Man WD, Nolan CM. Mood disorder in idiopathic pulmonary fibrosis: response to pulmonary rehabilitation. ERJ Open Res. 2023 May 22;9(3):00585-2022. doi: 10.1183/23120541.00585-2022. PMID: 37228278; PMCID: PMC10204825.]

Response 1. Thank you for taking time to search and suggest articles for us. We introduced two of the articles you suggested. [Besides, even though anxiety and depression symptoms are present in 35% respectively 37% of IPF patients, less than 5% were prescribed medication for mood disorders [96], indicating they are underrecognized and undertreated. Even tough antifibrotic medi-cation prescribed for patients with IPF and for progressive ILDs, is not an antipsychotic treatment, by improving dyspnea, cough and exercise capacity, can reduce the incidence and severity of anxiety and depression [97]] Page 18, line 635-640

Comments 2.  I found the paper interesting and well written; I suggest modifying the title as the paper does not focus properly and only on neurological implication of fibrosing interstitial lung disease but mostly on neurocognitive and neuropsychiatric impairment. 

Response 2. Thank you for pointing this out. At your suggestion, we changed the article name from “Neurological Implications of Fibrosing Interstitial Lung Diseases” to “Neurocognitive and neuropsychiatric implications of Fibrosing Interstitial Lung Diseases

Comments 3.  Further studies are needed to proper specify for each disease the most appropriate treatment and support required

Response 3. Thank you for your suggestion. We introduced a discussion section where we briefly presented the interventions and rehabilitation methods, and future directions. (Page 18, line 641-653) [ Moreover, there is also a complete lack of data regarding neurologic rehabilitation programs for patients with ILDs. Future research should concentrate on the immediate determinants of neurological impairment and on the most feasible interventions aimed to prevent neurocognitive morbidity. They may provide significant insights into the detection, natural progression, prognosis, and underlying mechanisms of neuro-physiological deficits. This will guide the development, implementation, and fine-tuning of intervention programs.

Increased identification of neurocognitive impairments may benefit patients by raising physician awareness and potentially leading to increased referrals to rehabilitation specialists, neuropsychologists, speech and language therapists, and other healthcare providers who can provide interventions such as cognitive remediation.]

Round 2

Reviewer 2 Report

Comments and Suggestions for Authors

Thanks for the editor's invitation to review the manuscript.

Congratulations to the authors, they adequately answered the comments and performed the required edits.